# Subjective Quality of Life and Its Associations among First Episode Psychosis Patients in Singapore

**DOI:** 10.3390/ijerph17010260

**Published:** 2019-12-30

**Authors:** Pratika Satghare, Edimansyah Abdin, Shazana Shahwan, Boon Yiang Chua, Lye Yin Poon, Siow Ann Chong, Mythily Subramaniam

**Affiliations:** 1Research Division, Institute of Mental Health, Buangkok Green Medical Park, 10 Buangkok View, Singapore 539747, Singapore; edimansyah_abdin@imh.com.sg (E.A.); Shazana_MOHAMED_SHAHWAN@imh.com.sg (S.S.); boon_yiang_chua@imh.com.sg (B.Y.C.); siow_ann_chong@imh.com.sg (S.A.C.); mythily@imh.com.sg (M.S.); 2Department of Early Psychosis Intervention, Institute of Mental Health, Singapore 539747, Singapore; lye_yin_poon@imh.com.sg

**Keywords:** first episode psychosis, quality of life, duration of untreated psychosis

## Abstract

**Background**—Mental disorders have been found to affect quality of life (QOL) in patients. The current study aimed to determine QOL among first episode psychosis (FEP) patients and explore its associations with sociodemographic as well as clinical factors. **Methods**—Data for this study were collected as a part of an Early Psychosis Intervention Program (EPIP)-Smoking and Alcohol use survey. At baseline, 280 outpatients aged 15–40 years old diagnosed with FEP, with no prior or minimal treatment, no history of medical or neurological disorder, and no history of substance abuse, were recruited. Sociodemographic details, diagnosis, length of duration of untreated psychosis (DUP), and World Health Organization Quality of Life assessment—abbreviated version (WHOQOL-BREF) scores were obtained. **Results**—After adjusting for all covariates, older age (*p* = 0.036), females, and participants diagnosed with brief psychotic disorder (*p* = 0.04) were associated positively, whereas separated/divorced participants, those with lower education, unemployed (*p* = 0.01), and longer DUP were seen to be negatively associated with different domains of QOL. **Conclusion**—Higher WHOQOL-BREF scores denote better QOL. Overall, female participants as compared to male participants and those diagnosed with brief psychotic disorder in this sample reported better QOL.

## 1. Background

Quality of life (QOL) is a multidimensional concept comprising physical, social, emotional, productive, and material wellbeing which can be assessed both subjectively and objectively [1]. According to the World Health Organization (WHO), quality of life is defined as “the individual’s perception of their position in life in the context of the culture and value systems in which they live and in relation to their goals” [2]. QOL is a key outcome measure in the planning and evaluation of health services and policies for various chronic disorders including mental illnesses [3]. In the field of healthcare, QOL is measured in terms of the impact of health status on the well-being (physical, emotional and social) of an individual [4]. Prior cross-sectional and comparative studies among patients with mental illnesses such as schizophrenia, major depressive disorder, and bipolar disorder have found mental illnesses to be associated with poor QOL [1,5,6]. Empirical findings from research studies among patients with mental illness have shown depression, positive psychotic symptoms, unmet basic, social, and functioning needs, lower satisfaction with daily living, social adjustment, size of support network, finances, and verbal intelligence to be the strong predictors of subjective quality of life [7,8,9,10]. Measuring QOL among people with mental disorders is crucial given the main emphasis of mental health services on symptom remission and in improving functional ability.

Psychotic disorders are severely debilitating illnesses of the young, with their onset and maximum impact occurring during the sensitive developmental adolescent period and early adult life [11]. Patients with psychosis display poor social skills, negative outcomes, social cognitive deficits, and poor functioning [12,13]. Dissatisfaction owing to deteriorated functioning and poor QOL among patients with psychosis is attributed to neurocognitive deficits along with impairment in theory of mind that has negative effects on patients’ clinical insight about their existing mental illness [14]. Prior literature has found strong relationships among negative symptoms, psychopathology, and QOL of patients during the early course of schizophrenia [15,16,17]. First episode psychosis (FEP) is defined as the first time that an individual experiences a psychotic episode which, being unfamiliar and disturbing, causes confusion, distress, and trauma for the patient as well their families. Patients with FEP experience symptoms such as reduced concentration, decreased motivation, depressed mood, sleep disturbance, anxiety, suspiciousness, social withdrawal, deterioration in functioning, hallucinations, delusions, and confused thinking, resulting in poor QOL [18].

Previous research among an Asian sample of FEP patients reported lower EuroQOL5 Dimensions (EQ-5D) index representing poor health-related quality of life (HRQoL) at baseline as compared to the general population [19]. Among FEP patients, male gender, depression, comorbid personality disorder, greater positive and negative psychotic symptoms, higher caregiver burden, and deteriorated psychological wellbeing have been reported to be correlates of lower subjective QOL [17,20,21]. Longer duration of untreated psychosis (DUP) is associated with deficit in psychological QOL and predicts poor response to treatment as well as recovery leading to worse clinical, cognitive, and functional outcomes [22]. By contrast, a review of literature reported the effect of premorbid adjustment to be prominent on QOL in FEP patients as compared to the effect of DUP [23,24]. 

Singapore is a multiethnic country situated in Southeast Asia with a total population of approximately 5.5 million, predominantly comprising people of Chinese, Malay, and Indian ethnicities [25]. Most of the extant research has been carried out among patients with chronic schizophrenia [16,26], and few studies have explored hazardous alcohol use [27], medication adherence [28], and pathways to care [29] among patients with FEP in Singapore. However, there is dearth of knowledge on the impact of first episode psychosis (FEP) on physical health, psychological health, social relationships, and environment domains of QOL in the multiracial population in Singapore. In view of this, the current study aimed to assess subjective QOL across the four domains of QOL and to elucidate its associations with sociodemographic and clinical variables at baseline among FEP patients seeking treatment in a tertiary care psychiatric hospital in Singapore.

## 2. Methods

This study was conducted among 280 consecutive outpatients with first episode psychosis (FEP) from the Early Psychosis Intervention Program (EPIP) at the Institute of Mental Health (IMH), Singapore. EPIP was launched in 2001 as a nationwide program at IMH to prevent unfavorable outcomes and improve clinical outcomes and QOL status of FEP patients [30]. The participants were recruited at baseline fulfilling the following criteria: (1) age between 15 and 40 years; (2) able to speak and understand English, and (3) FEP with no prior or minimal treatment, defined as <12 weeks antipsychotic medications (4) without current history of substance abuse and (5) no history of major medical or neurological illness, within 3 months of being included in EPIP. Patients recruited were identified as clinically stable to participate in the study according to their treating clinicians at the time of recruitment. Study team members were well versed in the informed consent process; study procedures were explained to all participants before seeking their consent for participating in the study. For participants below the age of 21 (the age of majority in Singapore), consent was obtained from a parent/legally acceptable representative. A written parental consent was documented on the informed consent form. On completion of the survey, participants were reimbursed with SGD 30 as an inconvenience fee. The study received ethics approval from National Healthcare Group Domain Specific Review Board. The current research study was performed in accordance with ethical standards of the National Healthcare Group Domain Specific Review Board, Singapore (ethical code No. 2013/01045) and with the 1964 Helsinki declaration and its later amendments or comparable ethical standards. Ethics approval and consent to participate: The study was approved by the National Healthcare Group Domain Specific Review Board in Singapore, and written informed consent was obtained from the participants. Availability of data and materials: Data supporting the findings are available upon request. Please contact the Principal Investigator of this study, Mythily Subramaniam (mythily@imh.com.sg), for data availability.

### 2.1. Measures

All the data except clinical measures were collected through a self-administered survey in the English language using an iPad. Further information was collected using the following questionnaires: 

(i)Sociodemographic questionnaire: The questionnaire collected data on age, gender, ethnicity, education level, marital and employment status, and personal/household income.(ii)Clinical Data: The data on diagnosis and duration of untreated psychosis (DUP) was collected from patient medical records. FEP patients were assessed by trained clinicians at baseline (within 3 months of enrolment in EPIP) to establish diagnosis using Structured Clinical interviews for Diagnostic and Statistical Manual of Mental Disorders, 4th edition (SCID-clinical version) (DSM-IV) [31]. DUP was calculated as time in months between onset of psychotic symptoms and initiation of treatment for psychosis.(iii)World Health Organization Quality of Life assessment—abbreviated version (WHOQOL-BREF) [32]: It is a 26-item, self-administered questionnaire, based on subjective evaluation, measuring an individual’s perception of QOL. The 26 items assess four domains related to QOL—physical health (activities of daily living, dependence on medical treatment, energy and fatigue, mobility, pain, discomfort, work capacity, sleep), psychological health (bodily image and appearance, negative and positive feelings, self-esteem, spirituality, concentration), social relationships (personal relationships, social support), and environment (finances, physical safety, access to health services, home environment, opportunities to acquire new information, leisure activities, physical environment, transport). The questionnaire also comprises 2 items which are based on the individual’s overall perception of QOL and general health, which are not included in the scoring. All items are constructed on variations of a 5-point Likert scale, with scores from 1 to 5, enquiring on how the individual felt, with the responses being ‘how much’, ‘how completely’, ‘how often’, ‘how good’ or ‘how satisfied’. Calculation of the scores for the 4 domains was done by taking the mean of all items from the domain, multiplying by 4, and then transforming the value obtained to a 4–20 scale. Higher scores denote higher QOL except for items 3, 4, and 26, which were reverse-scored. WHOQOL is a reliable, well-validated instrument [33] and shown to have good internal consistency for total and individual domain scores among patients with schizophrenia [34]. Sim et al.’s study has examined this instrument to assess QOL among patients with schizophrenia and co-morbid depression in Singapore [35]. Good construct validity and internal consistency was evidenced by high Cronbach’s alpha coefficients for the domains of physical health (0.79), psychological health (0.82), social relationships (0.81), and environment (0.83) [36].

### 2.2. Statistical Analysis

All analyses were conducted using SAS 9.2. Descriptive statistics were computed for the basic sociodemographic and clinical variables. Mean and standard deviations (SDs) were calculated for continuous variables and frequencies and percentages for categorical variables. Sociodemographic and clinical correlates of 4 domains of QOL were determined by multiple regression analysis. All 4 domains of QOL were treated as dependent variables, and age, sex, ethnicity, marital status, work status, education, diagnosis, and DUP were included as independent variables in the regression model. Significant associations were set at *p* < 0.05.

## 3. Results

The sociodemographic and clinical characteristics of the respondents are presented in Table 1. A total of 280 participants were recruited at baseline. The mean age of the participants enrolled in the study was 25.8 (SD = 6.2) years (Table 1). A total of 50.7% of the participants in the study sample were males. A total of 71.4% were Chinese, 15.7% were of Malay ethnicity, 11.4% were Indians, and 1.4% belonged to other ethnic groups. Most of the participants were single (n = 239, 85.4%), had completed polytechnic diploma (n = 65, 23.2%), and were employed (n = 118, 43.1%) at the time of assessment. In terms of diagnosis, 71.1% (n = 167) participants were diagnosed with schizophrenia spectrum disorders, 19.2% (n = 45) with brief psychotic disorder, and 9.8% (n = 23) were diagnosed with an affective disorder (Table 1). The mean score for duration of untreated psychosis (DUP) was 13.6 (SD = 21.7) months. Mean scores (ranging from 4 to 20) for the WHOQOL-BREF domains were observed to be 13.9 (physical health), 12.1 (psychological health), 12.8 (social relationships), and 13.2 (environment). The variance of quality of life explained by each of the significant independent factors included in the regression model was 16% variance of physical health and psychological domains, whereas 18% variance of social relationship and 21% variance of environment domain, respectively.

Table 2 shows the socio-demographic and clinical correlates of WHOQOL-BREF. After controlling for all covariates in multiple linear regression analyses, older age (β = 0.09, 95% CI = 0.01 to 0.17) (*p* = 0.036) was seen to be positively associated with the psychological health domain of QOL. Participants belonging to ‘Other’ ethnic group (β = 4.97, 95% CI = 1.27 to 8.66) (*p* = 0.009) were positively associated with the physical health domain of QOL as compared to those of Chinese ethnicity. Females participants were found to be positively associated with QOL on all the domains ((physical health: *p* = 0.038) (β = 0.74, 95% CI = 0.04 to 1.43), (psychological health: *p* = 0.020) (β = 0.96, 95% CI = 0.16 to 1.77), (social relationship: *p*= 0.022) (β = 0.942, 95% CI = 0.14 to 1.75), and (environment: *p* = 0.033) (β = 0.75, 95% CI = 0.06 to 1.43)). Separated/divorced participants were seen to be associated with poor QOL on physical health (β = −2.91, 95% CI = −5.17 to −0.64) (*p* = 0.012), psychological health (β = −3.25, 95% CI = −5.89 to −0.61) (*p* = 0.016), and social relationship (β = −1.33, 95% CI = −6.13 to 0.88) (*p* = 0.009) domains. Further, participants who were unemployed at the time of the study were seen to be poorly associated with QOL on the social relationships domain (β = −0.50, 95% CI = −2.28 to −0.33) (*p* = 0.009). Participants who had completed education up to primary school leaving examination (PSLE) level or below (β = −2.45, 95% CI = −4.49 to −0.41) (*p* = 0.019) and secondary level (β = −1.96, 95% CI = −3.74 to −0.19) (*p* = 0.03) were found to be associated with poor QOL on the environment domain. Participants who had completed ‘A’ level education were found to be associated with poor QOL on the social relationships domain (β = −1.70, 95% CI = −3.40 to −0.002) (*p* = 0.05). Participants diagnosed with brief psychotic disorder as compared to schizophrenia spectrum disorder (β = 1.13, 95% CI = 0.05 to 2.21) (*p* = 0.04) were positively associated with QOL on psychological health domain. Longer DUP was found to be negatively associated with physical health (β = −0.03, 95% CI = −0.04 to −0.01) (*p* = 0.002) and psychological health (β = −0.02, 95% CI = −0.04 to −0.003) (*p* = 0.023).

## 4. Discussion

The current study aimed to investigate the associations of subjective QOL among patients with FEP at baseline, i.e., within three months of being accepted by the EPIP, Singapore. QOL was seen to be significantly associated with sociodemographic variables, diagnosis, and DUP. The current study, with regards to age, revealed that FEP patients with older age reported better psychological QOL as compared to younger patients. This finding may be attributed to the fact that individuals belonging to the older age group may be more experienced in coping with their disorder symptoms, which may lower stress levels, something that has a positive impact on social and occupational functioning [37].

It is evident from prior literature that substantial gender differences exist among psychiatric population in terms of age of onset, clinical features, course and outcome, daily functioning, quality of life, and impairment [38,39]. The current study found that female participants were associated with better QOL as compared to males, which is in line with prior research [40,41]. Poor QOL can be attributed to early manifestation of disease, severe course, intellectual impairment, social deficits, disabling outcome of disease or the influence of sexual hormones among male patients with FEP leading to premorbid dysfunction, social deficits, and intellectual impairment as compared to females who manifest symptoms later in life with better response to treatment and higher compliance rates [42,43,44]. Our study found significant association between the physical health domain of QOL and the ‘other’ ethnic group versus the Chinese ethnic group, but it is difficult to interpret the finding, as this is a heterogeneous group.

Participants with FEP who were separated/divorced were negatively associated with QOL in all domains. This finding is in line with prior literature which suggests that the lack of marital communication and lack of an intimate relationship with a spouse can cause a decline in QOL attributed to loneliness, emotional turmoil, and depressed mood [45,46]. A study reported that married individuals experience better QOL due to the presence of intimate personal relationships, which lowers the level of the stress hormone, cortisol, as compared to those who are separated or divorced [47]. Evidence from prior population-based studies suggests that social support, family involvement, stress buffering effects, and feeling of belonging and purpose are the benefits of marriage that fulfil an individual’s psychological needs and thus improve QOL [48,49]. A few other studies have also reported that the effect of marital status on subjective QOL is moderated by culture, age, gender, and education [50,51]. 

Prior studies have proposed that education attained influences an individual’s social, psychological, and overall subjective wellbeing by developing their self-esteem, social skills, and socioeconomic conditions [52,53]. Further, it is evident that social skills and academic achievements consistently share a reciprocal relationship [54]. FEP patients with lower education attainment tend to report poorer QOL on the environment domain (including finances, leisure activities, and access to health services). This could possibly be linked to the patient’s employment status, as lower education attainment with poor academic and interpersonal skills limits the job opportunities for an individual, which may reduce engagement in leisure activities due to financial insecurity, leading to lower psychological wellbeing [55] and access to health services [56,57]. 

Employment, being an essential aspect of one’s social environment, is directly related to health and QOL. Among patients with mental illness, being employed enables financial independence and may aid in reducing symptoms by helping patients to cope more effectively with illness symptoms through healthy ways that improve daily functioning, which in turn may help to improve their QOL [58]. The current study found unemployed FEP patients to be negatively associated with the social relationships domain of QOL as compared to employed FEP patients. Unemployment is a stressful event that adversely impacts psychological and physical functioning, leading to anxiety, depression, somatization, and poor life satisfaction [59]. Gore et al. (1978) reported that social support moderates the consequences of unemployment and being unemployed exacerbates a low sense of social support and elevates stress [60]. Changes in various factors, such as self-esteem, self-efficacy, and stress-process-related factors such as emotional distress, somatic distress, personality traits, coping styles, and support from others, positively influences QOL among patients with schizophrenia as compared to changes in illness symptoms [61]. Psychosocial interventions such as counselling, psychoeducation workshops, and support groups to develop coping strategies carried out by various mental health professionals could provide emotional support, education about illness, effective communication skills, and practical assistance to re-establish a daily routine for patients as well as their family members to support their mentally ill family member [62].

QOL has emerged as a vital indicator of functional outcome in chronic and first episode psychosis (FEP) samples [63,64]. Participants diagnosed with brief psychotic disorders expressed a better QOL in the psychological health domain as compared to those diagnosed with schizophrenia spectrum disorders. This is attributed to the characteristics of brief psychotic disorder comprising an acute onset with a relatively short duration of illness, usually less than a month, followed by the return of normal functioning and usual level of activity, higher level of symptom awareness, and less symptom severity as compared to schizophrenia spectrum disorders [65]. Further, predominance of positive, negative, and cognitive symptoms in schizophrenia with mood disturbances and lack of insight [66] results in a significant decline in the psychological health domain among these patients, in contrast to brief psychotic disorder [67]. 

The study also found that among patients with FEP, longer DUP was negatively associated with QOL across all the domains, which is in line with prior research [35,68,69]. Untreated psychosis has harmful effects on the functioning of the brain resulting in cognitive and behavioral deficits steering the increase in negative symptoms [70] and lower remission rates over the course of illness as compared to those with a shorter DUP, where increased likelihood of symptom remission is observed [71,72]. However, a meta-analysis among patients with schizophrenia indicated that although longer DUP predicts symptom severity and deteriorated functioning, it is not associated with poor QOL [73].

The results of the study must be considered in view of some limitations. FEP participants were recruited from a tertiary care psychiatric hospital and may not be reflective of patients seeking care in the primary or secondary care settings. The results may not be representative of all FEP patient population seeking treatment under EPIP, as only clinically stable FEP patients were recruited. Further, only English-speaking patients were included in the self-administered survey, which could impose a sampling bias. Causal relationships cannot be established from the current research due to the cross-sectional design of the study. On the other hand, notwithstanding these limitations, the study was conducted among a large sample population of patients with FEP. All the questionnaires were self-administered, which lowers the risk of bias. The findings from this study address the gap in the existing literature in relation to QOL among multiethnic treatment-seeking FEP outpatients in Singapore. 

## 5. Conclusions

Subjective QOL is a valuable outcome assessment in treating patients with FEP, as early intervention and treatment of patients with psychotic symptoms may result in reduced morbidity, decreased suicide rate, improved QOL, and functional recovery. The current study adds to the new knowledge on the different facets of QOL in patients with FEP in terms of gender differences as well as the duration of untreated psychosis. As we are aware, the goal of scientific health research is to improve the health status of the population through the production of new evidence and knowledge, which can have an influence by improving mental health policies, helping to develop mental health strategies and programs, and assisting in clinical decision making. Our study outcomes can thus expand understanding in the field and help in designing strategies and planning interventions to improve the QOL among FEP patients.

## Figures and Tables

**Table 1 ijerph-17-00260-t001:** Socio-demographic and clinical characteristics of participants (n = 280).

	N	%	Mean	SD
Sex				
Female	138	49.3		
Male	142	50.7		
Ethnicity				
Chinese	200	71.4		
Malay	44	15.7		
Indian	32	11.4		
Others	4	1.4		
Marital Status				
Never Married	239	85.4		
Currently Married	34	12.1		
Separated/ Divorced	7	2.5		
Main Work Status (Last 12 months)				
Employed	118	43.1		
Economically inactive	78	28.5		
Unemployed	78	28.5		
Education				
PSLE and below	7	2.5		
Secondary	12	4.3		
O’/’N’ level	58	20.7		
A’ level	20	7.1		
Nitec/Higher Nitec	45	16.1		
Polytechnic Diploma	65	23.2		
Other Diploma and professional Qualification	22	7.9		
University	51	18.2		
Age			25.8	6.2
Diagnosis				
Schizophrenia spectrum disorder	167	71.1		
Affective disorder	23	9.8		
Brief psychotic disorder	45	19.2		
Duration of Untreated Psychosis (DUP) since onset of symptoms			13.6	21.7
QOL Domains				
Physical health			13.9	2.7
Psychological health			12.1	3.1
Social relationships			12.8	3
Environment			13.2	2.8
Total	280	100		

**Table 2 ijerph-17-00260-t002:** Sociodemographic and clinical correlates of WHOQOL-BREF at baseline.

	Physical Health				Psychological Health				Social Relationships				Environment			
	Beta Coefficient	95% CI		*p- Value*	Beta Coefficient	95% CI		*p*-*Value*	Beta Coefficient	95% CI		*p-Value*	Beta Coefficient	95% CI		*p-Value*
Age	0.04	−0.03	0.11	0.200	0.09	0.01	0.17	**0.036**	−0.03	−0.11	0.05	0.480	−0.01	−0.08	0.05	0.694
Sex																
Female	0.74	0.04	1.43	**0.038**	0.96	0.16	1.77	**0.020**	0.94	0.14	1.75	**0.022**	0.75	0.06	1.43	**0.033**
Male	0.00			.	0.00			.	0.00			.	0.00			.
Ethnicity																
Chinese	0.00			.	0.00			.	0.00			.	0.00			.
Malay	−0.53	−1.53	0.47	0.296	0.28	−0.88	1.45	0.634	0.31	−0.85	1.47	0.601	−0.41	−1.40	0.58	0.414
Indian	0.11	−0.94	1.16	0.833	1.19	−0.03	2.40	0.057	0.75	−0.46	1.96	0.225	0.17	−0.86	1.20	0.746
Others	4.97	1.27	8.66	**0.009**	−0.89	−4.39	4.21	0.968	−0.63	−4.91	3.66	0.774	3.26	−0.38	6.90	0.079
Marital Status																
Never Married	0.00			.	0.00			.	0.00			.	0.00			.
Currently Married	−0.41	−1.54	0.72	0.477	−0.41	−1.73	0.91	0.538	1.08	−0.23	2.39	0.106	0.29	−0.83	1.40	0.614
Separated/Divorced	−2.91	−5.17	−0.64	**0.012**	−3.25	−5.89	−0.61	**0.016**	−1.33	−6.13	−0.88	**0.009**	−1.86	−4.09	0.38	0.103
Main Work Status (Last 12 months)																
Employed	0.00			.	0.00			.	0.00			.	0.00			.
Economically inactive	0.25	−0.61	1.11	0.570	−0.70	−1.71	0.30	0.170	0.51	−1.15	0.85	0.770	0.39	−0.46	1.24	0.371
Unemployed	−0.08	−0.92	0.76	0.855	−0.75	−1.74	0.23	0.131	-0.50	−2.28	−0.33	**0.009**	−0.78	−1.61	0.05	0.067
Education																
PSLE and below	−1.42	−3.49	0.65	0.178	−2.00	−4.41	0.41	0.103	−2.04	−4.44	0.36	0.096	−2.45	−4.49	−0.41	**0.019**
Secondary	−1.28	−3.08	0.52	0.162	0.10	−1.99	2.20	0.922	−1.30	−3.38	0.79	0.222	−1.96	−3.74	−0.19	**0.030**
O’/’N’ level	0.11	−1.07	1.29	0.855	0.43	−0.94	1.80	0.538	−0.58	−1.94	0.79	0.406	−0.01	−1.17	1.15	0.989
A’ level	−0.19	−1.66	1.28	0.799	0.12	−1.59	1.82	0.894	−1.70	−3.40	−0.002	**0.050**	−0.58	−2.03	0.86	0.428
Nitec/Higher Nitec	−0.28	−1.50	0.93	0.646	0.60	−0.82	2.02	0.409	−0.42	−1.83	1.00	0.562	−1.11	−2.31	0.09	0.071
Polytechnic Diploma	0.19	−0.90	1.28	0.733	0.64	−0.63	1.91	0.321	−0.51	−1.78	0.75	0.423	−0.20	−1.28	0.87	0.709
Other Diploma and professional Qualification	−1.36	−2.83	0.11	0.069	−0.62	−2.33	1.10	0.479	−1.04	−2.74	0.66	0.230	−1.64	−3.09	−0.20	0.026
University	0.00			.	0.00			.	0.00			.	0.00			.
Diagnosis																
Schizophrenia spectrum disorder	0.00			.	0.00			.	0.00			.	0.00			.
Affective disorder	0.24	−0.94	1.43	0.688	0.54	−0.83	1.92	0.438	1.06	−0.31	2.43	0.128	1.08	−0.09	2.24	0.070
Brief psychotic disorder	0.10	−0.83	1.03	0.831	1.13	0.05	2.21	**0.040**	0.52	−0.55	1.59	0.338	0.41	−0.50	1.32	0.375
Duration of Untreated Psychosis (DUP) since onset of symptoms	−0.03	−0.04	−0.01	**0.002**	−0.02	−0.04	−0.003	**0.023**	0.003	−0.02	0.02	0.823	−0.01	−0.03	0.00	0.166

*-Significant results are shown in bold.*

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
