# Peer review of "Subjective Quality of Life and Its Associations among First Episode Psychosis Patients in Singapore"

_ijerph, 2019, doi:10.3390/ijerph17010260_

Round 1

Reviewer 1 Report

In this study the authors investigated perceived quality of life in out-patients in their first episode of psychosis. They also explore the role of many socio-demographic (e.g., age, gender) and clinical (e.g., duration of untreated psychosis) variables in affecting subjective quality of life among those patients.

In general, the article is interesting because it pays attention on a crucial aspect for individuals with mental illness. Quality of life could be seriously impaired in patients with psychosis which in turn could lead to a several other adverse outcomes, such as depressive disorder.

However, some specific comments and questions include:

1. In the introduction the text is fluent and well supported by literature. However, towards the end of this section, there is a sudden change of subject (line 73). I suggest including a better link between the introductory part, research question and aims of the study.

2. I don't see the importance of the results. What is the clinical and social implication or contributions of the findings from this article? For example, it is interesting to investigate the result concerning young people with lower perceived quality of life than older ones. Moreover, these people will rejoin society. How can these results help address the subsequent inclusion in society? Furthermore, the conclusions are rather brief, and this doesn’t help readers to understand more authors' points of view from this article.

Author Response

Respected Reviewer,

Thank you and appreciate your help in improving the quality of my manuscript with your valuable comments and suggestions. I have made the edits as per comments and provided the explanation wherever applicable.

In this study the authors investigated perceived quality of life in out-patients in their first episode of psychosis. They also explore the role of many socio-demographic (e.g., age, gender) and clinical (e.g., duration of untreated psychosis) variables in affecting subjective quality of life among those patients.

In general, the article is interesting because it pays attention on a crucial aspect for individuals with mental illness. Quality of life could be seriously impaired in patients with psychosis which in turn could lead to a several other adverse outcomes, such as depressive disorder.

However, some specific comments and questions include:

In the introduction the text is fluent and well supported by literature. However, towards the end of this section, there is a sudden change of subject (line 73). I suggest including a better link between the introductory part, research question and aims of the study.

-   Related edits done in the manuscript.

I don't see the importance of the results. What is the clinical and social implication or contributions of the findings from this article? For example, it is interesting to investigate the result concerning young people with lower perceived quality of life than older ones. Moreover, these people will rejoin society. How can these results help address the subsequent inclusion in society? Furthermore, the conclusions are rather brief, and this doesn’t help readers to understand more authors' points of view from this article.

Related edits done in the manuscript.

Thank you.

Kind Regards,

Pratika

Reviewer 2 Report

IJERP Reviewer Comment

Thank you the opportunity to review this manuscript. First, I would like to appreciate the authors for assessing the subjective QOL of FEP patients across four domains and elucidating its associations with socio-demographic and clinical variables at baseline among FEP patients seeking treatment in a tertiary care psychiatric hospital in Singapore. I have the following comments;

Background

Literature search showed that there is no dearth of knowledge about first-episode psychosis (FEP) in the multiracial population in Singapore. Authors should strengthen their background with reference to already existing literature about FEP patients in Singapore. Readers should be acquainted with the prevalence data about FEP in multiracial population in Singapore. Readers also need to be told what is already known about FEP patients in Singapore and thereafter, what is not known as you appear to be demonstrating.  E.g. of FEP patient literature in Singapore already published:

Tang C, Subramaniam M, Ng BT, Abdin E, Poon LY, Verma SK. Clozapine Use in First-Episode             Psychosis: The Singapore Early Psychosis Intervention Programme (EPIP) Perspective.     The Journal of clinical psychiatry. 2016 Nov;77(11):e1447-53.

Cetty L, Shahwan S, Satghare P, Devi F, Chua BY, Verma S, Lee H, Chong SA, Subramaniam M.             Hazardous alcohol use in a sample of first episode psychosis patients in Singapore. BMC psychiatry. 2019 Dec;19(1):91.

Tan C, Abdin E, Liang W, Poon LY, Poon NY, Verma S. Medication adherence in first-episode psychosis patients in Singapore. Early Interv Psychiatry. 2019 Aug;13(4):780-788. doi:          10.1111/eip.12559. Epub 2018 Mar 9.

Chesney, Edward Abdin, Edimansyah Poon, Lye Yin Subramaniam, Mythily Verma, Swapna.     Pathways to Care for Patients With First-Episode Psychosis in Singapore. The Journal of       nervous and mental disease January 2016; 204(4).

Tan XW, Shanwan S, Satghare P, Chua BY, Verma S, Tang YZ, Chong SA. Trends in Subjective             Quality of Life among Patients with First Episode Psychosis-a One Year Longitudinal      Study. Frontiers in psychiatry. 2019;10:53.

Method

In the method section, include a statement regarding Helsinki Declaration. Also, indicate whether you conducted assumptions test prior to running your chosen statistical analysis

Results

In page 4 of 13, line 153, you wrote “Females participants” instead of female participants.

Discussion

This report is not clear: page 4 of 13, lines 170-172.  You began by reporting your finding and ended the sentence with a citation. Could you please report the finding first, then relate it to prior literature in a more clearer structure?

With regards to age, the current study revealed FEP patients with older age reported better psychological QOL as compared to younger patients as they may be more experienced to cope with their disorder symptoms which may lower stress levels that has positive impact on social and occupational functioning (35).

I am not sure what you are trying to report here in lines 186-187, ………….. communication and intimate relationship with spouse can cause decline in QOL attributed…

Do you mean that intimate relationship would result in QOL decline? I suggest you check your source again to be sure. This sentence is contradictory to your next sentence in lines 189-190.

In line 190, population based is usually written as population-based. There should be an hyphen between the two words. Line 200, in depth write as in-depth. 

Lines 201-202 is unclear, …..due to which they may have less time to involve in 201 leisure and socializing activities leading to smaller social network

The limitations of the study should be broadened.

Add the mental health policy implications of your research findings. Authors need to additionally discuss the implications of their study results for clinicians, physician training and future researchers

Author Response

Respected Reviewer,

Thank you and appreciate your help in improving the quality of my manuscript with your valuable comments and suggestions. I have made the edits as per comments and provided the explanation wherever applicable.

Thank you, the opportunity, to review this manuscript. First, I would like to appreciate the authors for assessing the subjective QOL of FEP patients across four domains and elucidating its associations with socio-demographic and clinical variables at baseline among FEP patients seeking treatment in a tertiary care psychiatric hospital in Singapore. I have the following comments;

Literature search showed that there is no dearth of knowledge about first-episode psychosis (FEP) in the multiracial population in Singapore. Authors should strengthen their background with reference to already existing literature about FEP patients in Singapore. Readers should be acquainted with the prevalence data about FEP in multiracial population in Singapore. Readers also need to be told what is already known about FEP patients in Singapore and thereafter, what is not known as you appear to be demonstrating.  E.g. of FEP patient literature in Singapore already published.

Related edits done in the manuscript.

In the method section, include a statement regarding Helsinki Declaration. Also, indicate whether you conducted assumptions test prior to running your chosen statistical analysis

Included in the manuscript, ‘The current research study was performed in accordance with ethical standards of the National Healthcare Group Domain Specific Review Board, Singapore (ethical code no:2013/01045 ) and with the 1964 Helsinki declaration and its later amendments or comparable ethical standards.’

In page 4 of 13, line 153, you wrote “Females participants” instead of female participants.

-correction done in the manuscript.

This report is not clear: page 4 of 13, lines 170-172.  You began by reporting your finding and ended the sentence with a citation. Could you please report the finding first, then relate it to prior literature in a more clearer structure? With regards to age, the current study revealed FEP patients with older age reported better psychological QOL as compared to younger patients as they may be more experienced to cope with their disorder symptoms which may lower stress levels that has positive impact on social and occupational functioning (35).

The above suggested correction is made in the manuscript.

 I am not sure what you are trying to report here in lines 186-187, ………….. communication and intimate relationship with spouse can cause decline in QOL attributed…Do you mean that intimate relationship would result in QOL decline? I suggest you check your source again to be sure. This sentence is contradictory to your next sentence in lines 189-190.

In line 190, population based is usually written as population-based. There should be an hyphen between the two words. Line 200, in depth write as in-depth. 

Lines 201-202 is unclear,due to which they may have less time to involve in 201 leisure and socializing activities leading to smaller social network

Clarification and corrections made in the manuscript.

Add the mental health policy implications of your research findings. Authors need to additionally discuss the implications of their study results for clinicians, physician training and future researchers.

Related edits done in the manuscript.

Thank you.

Kind Regards,

Pratika

Reviewer 3 Report

All of the references in the Reference list should be adjusted to IJERPH standards

Something is wrong with the stat. analysis – what kind of regression did authors actually use?

What about Normality, Linearity, Homoscedasticity and (multi)collinearity and singularity?

Tolerance and Variance Inflation Factor (VIF) – not shown!

If we estimate only on the basis of CI, then

Table 2. – regardless p<0.05, for almost every parameters examined (except Age - Psychological), the authors can not claim the significance, because 95% CI varies from below 1 to above 1, or vice versa.

Therefore, this paper should either further elaborated or rejected

Author Response

Respected Reviewer,

Thank you and appreciate your help in improving the quality of my manuscript with your valuable comments and suggestions. I have made the edits as per comments and provided the explanation wherever applicable.

All of the references in the Reference list should be adjusted to IJERPH standards

Correction done in the manuscript.

Something is wrong with the stat. analysis – what kind of regression did authors actually use?What about Normality, Linearity, Homoscedasticity and (multi)collinearity and singularity? Variance Inflation Factor (VIF) – not shown!

Information included to the manuscript.

If we estimate only on the basis of CI, thenTable 2. – regardless p<0.05, for almost every parameters examined (except Age - Psychological), the authors can not claim the significance, because 95% CI varies from below 1 to above 1, or vice versa.

Relevant changes done in the manuscript.  

Thank you.

Kind Regards,

Pratika

Reviewer 4 Report

This reviewer appreciates the efforts of the authors for their study title, " Subjective Quality of Life and its associations among First Episode Psychosis Patients in Singapore."

The following comments, feedback, and questions need to be addressed:

Background

     The authors need to include the epidemiology of mental disorders in Singapore, including prevalence trends and briefly the state of mental health care delivery system in Singapore. The ultimate goal of scientific health research is to improve population health status through the production of new evidence (new information and/or knowledge) that can influence policy, strategy, or clinical decision making.  The inclusion of existing evidence on such matters on the background section will serve as a launchpad for the discovery of novel evidence that can expand our understanding in the field.

Method

   What is the design of the study? It looks like a mix of a retrospective and a cross-sectional study. The study makes use of retrospective clinical data and a cross-sectional survey data. The  authors need to clearly state the design of the study.

 What exactly is/are the research questions? What is new information that this research is trying to find out? What knowledge gap is it trying to bridge?

What is the validity and reliability of the survey instruments?  Please state in quantitative terms.

Results and Discussion

What is the variance of quality of life explained  by each of the significant independent variables for each domain of quality of life or outcome variable?  Physical, Psychological, Social, and Environmental.  

Conclusion

The authors need to provide a coherent evidence-based narrative that adds new information and new knowledge in the field. The authors need to state how the research fills an identified knowledge gap and how the new knowledge will be used to improve quality of life through improved mental health policies and mental health programs in the study area and beyond.

Author Response

Respected Reviewer,

Thank you and appreciate your help in improving the quality of my manuscript with your valuable comments and suggestions. I have made the edits as per comments and provided the explanation wherever applicable.

The authors need to include the epidemiology of mental disorders in Singapore, including prevalence trends and briefly the state of mental health care delivery system in Singapore. The ultimate goal of scientific health research is to improve population health status through the production of new evidence (new information and/or knowledge) that can influence policy, strategy, or clinical decision making.  The inclusion of existing evidence on such matters on the background section will serve as a launchpad for the discovery of novel evidence that can expand our understanding in the field.

  What is the design of the study? It looks like a mix of a retrospective and a cross-sectional study. The study makes use of retrospective clinical data and a cross-sectional survey data. The  authors need to clearly state the design of the study.

The current study design is cross sectional in nature. The current paper reports data that was extracted from a longitudinal study, investigating smoking and alcohol use amongst outpatients who were seeking treatment at a tertiary level psychiatric hospital…Clarification stated in the manuscript.

What exactly is/are the research questions? What is new information that this research is trying to find out? What knowledge gap is it trying to bridge?

The current study aims to assess subjective QOL across the four domains and to elucidate its associations with socio-demographic and clinical variables at baseline among FEP patients seeking treatment in a tertiary care psychiatric hospital in Singapore.

What is the validity and reliability of the survey instruments?  Please state in quantitative terms.

The validity and reliability of WHOQOL-BREF, that was used as survey instrument for current study, is mentioned in the manuscript under methodology section.

What is the variance of quality of life explained  by each of the significant independent variables for each domain of quality of life or outcome variable?  Physical, Psychological, Social, and Environmental.  

The variance of Quality of life, R2 values, explained by each of the significant independent variables for each domain of quality of life, that is , physical, psychological, social and environmental are noted in the manuscript.

The authors need to provide a coherent evidence-based narrative that adds new information and new knowledge in the field. The authors need to state how the research fills an identified knowledge gap and how the new knowledge will be used to improve quality of life through improved mental health policies and mental health programs in the study area and beyond.

Related edits made in the manuscript.

Thank you.

Kind Regards,

Pratika

Round 2

Reviewer 1 Report

I am satisfied with authors' progresses made on the manuscript.

Author Response

Respected Reviewers,

Thank you for your comments.

Kind Regards,

Pratika

Reviewer 2 Report

I am very satisfied with the author's revision.

Author Response

(The authors gave the same response as above.)

Reviewer 3 Report

β coefficients should be presented with corresponding CIs in “Results” for each of independent variable.

If a CI includes 0, the association between the dependent variable (after adjusting for covariates) and the independent variable is not significant.

Education - Social relationships - A' level – β=-1.70, 95% CI=-3.40 to 0.00.

Duration of Untreated Psychosis (DUP) since onset of symptoms - Psychological health - β=-0.02, 95% CI=-0.04 to 0.00.

Some of the analysis results were also very questionable, for example:

DUP - Physical Health - β=-0.03, 95% CI=-0.04 to -0.00.

Age - Psychological Health - β=0.09, 95% CI=-0.01 to 0.17.

(That is why asked for Tolerance and Variance Inflation Factor (VIF) to be also shown.)

Also, about Ethnicity, only 4 respondents were “some other ethnicity” (nor Chinese, Malay or Indian) - so, what does the following finding actually tell us: “Participants belonging to ‘Other’ ethnic group (p value= 0.009) were negatively associated with physical health domain of QOL as compared to those of Chinese ethnicity.” (lines 149-150) ?

Author Response

Respected Reviewers,

Thank you for your comments. We acknowledge and have done the revisions as suggested.

Comments:

β coefficients should be presented with corresponding CIs in “Results” for each of independent variable. If a CI includes 0, the association between the dependent variable (after adjusting for covariates) and the independent variable is not significant.

Education - Social relationships - A' level – β=-1.70, 95% CI=-3.40 to - 0.002

Duration of Untreated Psychosis (DUP) since onset of symptoms - Psychological health - β=-0.02, 95% CI=-0.04 to - 0.003

β coefficients are included with corresponding CIs in “Results” for each of independent variable. Have done the corrections in the manuscript (Tables) accordingly and included three decimal values as we got no values (CI includes 0 as referred by the reviewer) after conversion to two decimal points. The corrected values confirm the association between the dependent and independent variable being significant.
DUP - Physical Health - β=-0.03, 95% CI=- 0.04 to - 0.01, Age - Psychological Health - β=0.09, 95% CI= 0.01 to 0.17. We have done the corrections and also included the variance factors in the results section. Also, about Ethnicity, only 4 respondents were “some other ethnicity” (nor Chinese, Malay or Indian) - so, what does the following finding actually tell us: “Participants belonging to ‘Other’ ethnic group (p value= 0.009) were negatively associated with physical health domain of QOL as compared to those of Chinese ethnicity.” (lines 149-150) ? For the ‘Other ethnic group’, we acknowledge it as a limitation due to its small sample size. The ethnic group that was classified as other ethnic group included only 4 respondents belonging to diverse and heterogenous ethnic origins. As such, we could not make any meaningful inferences and the results related to this group will not be discussed further in the paper.

Thank you and appreciate!

Kind Regards,

Pratika

Reviewer 4 Report

The authors have provided a point-to-point-response to the main comments provided and questions raised by this reviewer.  The manuscript may need further discretionary reviews by the authors. 

Author Response

(The authors gave the same response as above.)
